# Phenolic Profile, Antioxidant and Enzyme Inhibition Properties of the Chilean Endemic Plant *Ovidia pillopillo* (Gay) Meissner (Thymelaeaceae)

**DOI:** 10.3390/metabo12020090

**Published:** 2022-01-18

**Authors:** Carmen Cortés, Diego A. González-Cabrera, Ruth Barrientos, Claudio Parra, Javier Romero-Parra, Mariano Walter Pertino, Carlos Areche, Beatriz Sepúlveda, Jorge Bórquez, Alfredo Torres-Benítez, Mario J. Simirgiotis

**Affiliations:** 1Instituto de Farmacia, Facultad de Ciencias, Campus Isla Teja, Universidad Austral de Chile, Valdivia 5090000, Chile; carmenc1012@gmail.com (C.C.); diego.gonzalez@alumnos.uach.cl (D.A.G.-C.); ruth.barrientos@alumnos.uach.cl (R.B.); aljotobe19@hotmail.com (A.T.-B.); 2Laboratorio de Química Orgánica y Productos Naturales, Facultad de Ciencias Agronómicas, Universidad de Tarapacá, Av. General Velásquez 1775, Arica 1000000, Chile; cparra@uta.cl; 3Departamento de Química Orgánica y Fisicoquímica, Facultad de Ciencias Químicas y Farmacéuticas, Universidad de Chile, Olivos 1007, Casilla 233, Santiago 6640022, Chile; javier.romero@ciq.uchile.cl; 4Laboratorio de Química de Productos Naturales, Instituto de Química de Recursos Naturales, Universidad de Talca, Talca 3460000, Chile; 5Departamento de Química, Facultad de Ciencias, Universidad de Chile, Las Palmeras 3425, Nuñoa, Santiago 7800024, Chile; areche@uchile.cl; 6Departamento de Ciencias Químicas, Universidad Andres Bello, Campus Viña del Mar, Quillota 980, Viña del Mar 2520000, Chile; bsepulveda@uc.cl; 7Departamento de Química, Facultad de Ciencias Básicas, Universidad de Antofagasta, Antofagasta 1240000, Chile; jorge.borquez@uantof.cl; 8Center for Interdisciplinary Studies on the Nervous System (CISNe), Universidad Austral de Chile, Valdivia 5090000, Chile

**Keywords:** toxic plants, daphnetin, coumarins, glycosyl flavonoids, cholinesterase inhibition, UHPLC–PDA–OT-MS/MS analysis, antioxidants, endemic species

## Abstract

*Ovidia pillopillo* (Lloime) is an endemic species of the Valdivian Forest of Chile. Little is known on the chemistry and biological activity of this plant. In this study, the phenolic profile, antioxidant capacities and enzyme inhibition capacities (against tyrosinase and cholinesterase) of the plant were investigated for the first time. The phenolic profile of the plant was obtained by UHPLC-MS fingerprinting with high resolution, which showed the presence of several flavonoids and coumarins. The antioxidant potential was measured by FRAP and ORAC (45.56 ± 1.32; 25.33 ± 1.2 μmol Trolox equivalents/g dry plant, respectively) plus ABTS and DPPH methods (IC_50_ = 9.95 ± 0.05 and 6.65 ± 0.5 μg/mL, respectively). Moreover, the flavonoid and phenolic contents were determined (57.33 ± 0.82 and 38.42 ± 1.32, μg of Trolox and quercetin equivalents/100 g dry weight, respectively). The ethanolic extract showed cholinesterase (IC_50_ = 1.94 ± 0.07 and 2.73 ± 0.05 μg/mL, for AChE and BuChE, respectively) and tyrosinase (4.92 ± 0.05 μg/mL) enzyme inhibition activities. Based on these in vitro studies, in silico simulations were performed, which determined that the major compounds as ligands likely docked in the receptors of the enzymes. These results suggest that *Ovidia pillopillo* produce interesting special coumarins and flavonoids, which are potential candidates for the exploration and preparation of new medicines.

## 1. Introduction

*Ovidia pillopillo*, local name Lloime (Synonyms: *Daphne pillopillo* C. Gay), is an endemic plant from Valdivia’s temperate forest that is very purgative and emetic (Figure 1). The plant is a dioecious shrub with upright and bushy branches and gray bark that grows up to 700 m above sea level and reaches a height of 7 m. It bears white and scented flowers that appear in terminal or apical fascicles, and the leaves are alternating, subcoriaceous, oblong and elliptical evergreen leaves of 2 to 8 cm long [1]. The plant’s bark, fruits and leaves are all therapeutic (generally as vomitive, for constipation and treatment of syphilis), but in high doses are toxic [1,2].The plant has been employed since the ancient Mapuche pre-Columbian era, it was thrown into the sea and river waters to numb and catch fish. The infusion from the bark causes diarrhea, but the infusion from the same flowers work as an antidote. Extracts of this plant showed antifungal activity against the pathogenic soil fungi *Rhizoctonia solani* [3]. The toxic activity of the plant is attributed to daphnin (daphnetin 7-β-D-glucopyranoside) [1], a toxic coumarin constituent of the Daphne species. Moreover, daphnetin (7,8-dihydroxycoumarin), a major bioactive component extracted from the genus Daphne, was devoid of toxicity and showed antimicrobial properties [4,5]. Those interesting coumarins serve as chemotaxonomical markers of *Daphne* and *Ovidia* species [6,7].

To date, no UHPLC fingerprinting of the phenolic metabolites contained in Lloime (*O. pillopillo*) have been reported. As far as we know, there are also no scientific reports concerning to the antioxidant and enzyme inhibition potential of extracts from the plant. Our group recently investigated the enzyme inhibitory properties plus the antioxidant and chemical fingerprinting of several Chilean native plants and fruits [8,9,10,11,12]. Following our studies on South American plants, we now report the phenolic fingerprinting from ethanolic extracts of Lloime leaves by UHPLC-MS analysis, antioxidant activity and enzyme inhibitory potential (against cholinesterase and tyrosinase) for the first time.

## 2. Results

### 2.1. UHPLC-OT-MS Analysis of Lloime (O. pillopillo)

An ultra-high resolution liquid chromatography Orbitrap MS analysis UHPLC–PDA–OT-MS/MS combining full MS spectra and MS^n^ experiments revealed the presence of thirty-two compounds in Lloime, all of them reported for the first time. The composition of the extracts included phenolics acids, fatty acids, and several characteristic coumarins. Some of these compounds were identified by spiking experiments with available standards (Table 1). The analyses were confirmed using MS/MS data and comparing the MS fragments with those from the available bibliography. Figure 2 shows an UHPLC-DAD-MS chromatograms of Lloime from the Valdivian Forest, Chile. The structures of some of these compounds are depicted in Figure 3. Appendix A: Shows Full MS spectra and structures of several compounds detected in *O. pillo- pillo.*

Below the comprehensive metabolome analyses is explained.

#### 2.1.1. Coumarins

The prominent peak 4 with UV max at 227–290sh–325 nm a pseudomolecular ion at *m*/*z*: 353.08923 and diagnostic ion at *m*/*z*: 191.05633 (daphnetin 8-methyl ether) was identified as the methyl derivative of the toxic coumarin daphnin [1], identified as daphnin 8-O-methyl ether (daphnetin 7 O-glucose, 8-methyl ether, C_16_H_17_O_9__^−^_). In the same manner, peak 6 was identified as its methyl derivative daphnin 8-O-methyl ether-6¨-O-methyl-glucose (C_17_H_19_O_9__^−^_), and peak 7 as its isomer daphnetin 7 O-(5¨-O-methyl-glucose 8-O-methyl ether. Peak 25 was identified as the dicoumarin daphnoretin (C_19_H_12_O_7__^−^_) [6].

#### 2.1.2. Phenolic Acids

Peak 3 was identified as vanillic acid (C_8_H_8_O_4__^−^_), and peak 5 ion at *m*/*z*: 385.35066 as sinapoyl glucose (C_17_H_21_O_10__^−^_).

#### 2.1.3. Flavonoids

Peak 8 showing UV vis data of flavonol (254–354 nm) and a parent full MS ion at *m*/*z*: 579.12446 and daughter quercetin ion at *m*/*z*: 301.03524 was identified as quercetin 3-O-xylosyl-(1-2)-rhamnoside (C_26_H_27_O_15__^−^_) [13]; in the same manner, peak 14 was tentatively identified as the antiinflammatory quercetin 3-O-β-D-glucuronide (C_21_H_18_O_13__^−^_) [14], peak 18 as its acetyl derivative quercetin-3-O-β-D (2″-acetyl glucuronide (C_23_H_19_O_14__^−^_), peak 16 as its methyl derivative quercetin 3-O-β-D-glucuronide-methyl esther (C_21_H_17_O_12__^−^_) and peak 21 as quercetin 3-O-(2″-O-acetyl-glucuronide methyl ester), with the formula: C_24_H_21_O_14__^−^_ and peak 27 as quercetin. On the other hand, several compounds were identified as kaempferol derivatives (Uv max 264–347 nm). Peak 9 with a parent full MS ion at *m*/*z*: 593.15356 was identified as the lipid peroxidantion inhibitor kaempferol 3-O-rutinose (C_27_H_29_O_15__^−^_) [15] with a diagnostic kaemperol ion at *m*/*z*: 285.03991, and its isomer peak 18 as kaempferol-3-O-neohesperidose. Peak 10 was identified as kaempferol 3-O-β-D-glucose (C_21_H_19_O_11__^−^_), peak 13 as the immunostimulatory kaempferitrin (C_27_H_29_O_14__^−^_) [16], peak 22 as kaempferol 3-O-β-(2′-O-acetyl-β-D-glucuronide), with a formula: C_23_H_19_O_13__^−^_, peak 23 as kaempferol 3-O-glucuronide methyl ether (C_22_H_19_O_12__^−^_), and peak 26 as kaempferide (C_16_H_11_O_6__^−^_). Peak 16 showed UV max of luteolin (270–338 nm) and showed a deprotonated molecule at *m*/*z*: 461.07254 and was identified as the anti-inflammatory and antioxidant compound luteolin-7-O-glucuronide (C_21_H_17_O_12__^−^_) [17]. Similarly peak 21 showed a deprotonated molecule at *m*/*z*: 503.08521 and was identified as luteolin 3-O-(β-D-2’O-acetyl glucuronide (C_23_H_19_O_13__^−^_), with a luteolin fragment at *m*/*z*: 285.03993, peak 24 as luteolin 7-O-(β-D-2¨O-acetylglucuronide methyl ester (C_24_H_21_O_13__^−^_). Peak 19 with a UV max of flavanone (280 nm) was identified as 3,8″-binaringenin (C_30_H_23_O_10__^−^_), peak 11 as proanthocyanidin geranin B (C_30_H_23_O_11__^−^_), peak 12 as isoscoparine (C_22_H_21_O_11__^−^_) and, finally, peak 28 as acacetin (C_16_H_11_O_5__^−^_).

#### 2.1.4. Fatty Acids

Peak 30 with a pseudomolecular ion at *m*/*z*: 329.23346 was tentatively identified as 3,5,7-trihydroxyoleic acid (C_18_H_33_O_5__^−^_), peak 29 as hydroxyoctadecaenoic acid (C_18_H_35_O_3__^−^_) [18], peak 31 as hexadecatrienoic acid (C_16_H_25_O_2__^−^_) and peak 32 as 2-hydroxypalmitate (C_16_H_31_O_3__^−^_).

#### 2.1.5. Other Compounds

Peaks 1 was identified as quinic acid (C_7_H_11_O_6__^−^_) and peak 2 as isocitric acid (C_6_H_7_O_7__^−^_).

### 2.2. Antioxidant and Enzyme Inhibitory Capacities of Lloime

Lloime leaves extracts were assessed in vitro for cholinesterase and tyrosinase inhibitory potential. As far as we know, no previous reports regarding anti-enzymatic potential have been published for this species. The results are summarized in Table 2 and are expressed as IC_50_ values (µg/mL). Besides, the antioxidant potential of the ethanolic extract of this plant was measured by several methods, namely FRAP and ORAC (45.56 ± 1.32, 25.33 ± 1.2 μmol Trolox equivalents/g dry plant, respectively), plus ABTS and DPPH methods (IC_50_ = 9.95 ± 0.05, and 6.65 ± 0.5 μg/mL). This was complemented with the measurements of flavonoid and phenolic contents (57.33 ± 0.82 and 38.42 ± 1.32, μg of trolox and quercetin equivalents/100 g dry weight, respectively).

### 2.3. Docking Assays of Most Abundant Compounds

All compounds that turned out to be the most abundant in Lloime leaves extract (Figure 4) according to the UHPLC chromatogram (Figure 2) obtained from *O. pillopillo* leaves extract, as well as the known cholinesterase and tyrosinase inhibitors, galantamine and kojic acid respectively, were subjected to docking assays into the acetylcholinesterase catalytic site, the butyrylcholinesterase catalytic site and the tyrosinase catalytic site, in order to rationalize their inhibitory properties through the analysis of the protein molecular interactions in the light of the experimental inhibition activities obtained (Table 2). The best docking binding energies expressed in kcal/mol of each compound are shown in Table 3.

#### 2.3.1. Acetylcholinesterase (*Tc*AChE) Docking Results

Table 3 shows binding energies of 5-hydroxy-7-methoxy-2-phenyl-4*H*-chromen-4-one, luteolin 7,4′-dimethyl ether, Apigenin 5-glucoside, quercetin 3-O-ß-D-2″-acetylglucuronide, quercetin 3-O-ß-D-2″-acetylglucuronide methyl ester, quercetin 3-O-(ß-D-glucuronide) and quercetin 3-O-ß-D-(glucuronide methyl ester). Better energy parameters for Apigenin 5-glucoside, quercetin 3-O-ß-D-2″-acetylglucuronide, quercetin 3-O-ß-D-2″-acetylglucuronide methyl ester, quercetin 3-O-(ß-D-glucuronide) and quercetin 3-O-ß-D-(glucuronide methyl ester) were obtained than with standard galantamine. Although the aforementioned compounds possess slightly better energy profiles than the known inhibitor galantamine, the fact that the *Ovidia pillopillo* leaves extract contain many other active compounds, including the seven that we studied by docking assays, lead into a competition among all of them for the acetylcholinesterase catalytic site; not a refined or accurate result can be obtained as if an inhibition assay were done with each one of the derivatives separately. Nonetheless, it is clear that the good energies presented by these major compounds obtained from the extract are responsible for the strong inhibitory effect exerted over the enzyme.

5-Hydroxy-7-methoxy-2-phenyl-4*H*-chromen-4-one and luteolin 7,4′-dimethyl ether are arranged in the same manner into the acetylcholinesterase catalytic site, which could probably explain the similar binding energy descriptors between them (−9.154 kcal/mol and −10.506 kcal/mol, respectively). Likewise, both of these compounds are also settled similarly to quercetin 3-O-ß-D-2″-acetylglucuronide, quercetin 3-O-ß-D-2″-acetylglucuronide methyl ester and quercetin 3-O-ß-D-(glucuronide methyl ester) in terms of geometric molecular space. Nonetheless, probably due the lack of glycoside moities in 5-hydroxy-7-methoxy-2-phenyl-4*H*-chromen-4-one and Luteolin 7,4′-dimethyl ether, the obtained binding energies and, consequently, the inhibitory activities, are lower than for the other derivatives (Table 3 and Figure 5). The 5-hydroxy-7-methoxy-2-phenyl-4*H*-chromen-4-one formed one hydrogen bond with Tyr130 and two π-π interactions through the phenyl moiety at position 2- of the 4*H*-chromen-4-one and the amino acids Phe330 and Tyr334. On the other hand, luteolin 7,4′-dimethyl ether carried out two hydrogen bond interactions with Tyr121 and Tyr130, as well as a π-π interaction between Trp84 and the 4*H*-chromen-4-one core. The extra interaction with Tyr121 would explain the better binding energy profile of this derivative compared to 5-hydroxy-7-methoxy-2-phenyl-4*H*-chromen-4-one (Table 3).

Apigenin 5-glucoside shows a particular pattern of location into the catalytic site, and no similarities in terms of spatial arrangement were found with any of the other tested derivatives. Apigenin 5-glucoside displayed four hydrogen bond interactions with Asp72, Tyr121 and Tyr130. Besides, it can be noted that three π-π interactions with Trp84 and Phe330 residues were performed by this compound (Figure 5C).

Quercetin 3-O-ß-D-2″-acetylglucuronide, quercetin 3-O-ß-D-2″-acetylglucuronide methyl ester, and quercetin 3-O-ß-D-(glucuronide methyl ester) are arranged in similar ways, but with slight differences. Therefore, these derivatives shared some interactions with the same amino acids of acetylcholinesterase catalytic. In this sense, all three compounds, through their hydroxyl (-OH) groups at position 7- of the 4*H*-chromen-4-one framework, performed hydrogen bond interactions with His440. Moreover, the three of them showed π-π or T-shaped interactions with Phe330 and Phe331 accordingly, except for quercetin 3-O-ß-D-2″-acetylglucuronide methyl ester, which also carried out this sort of interaction with Phe290. Moreover, quercetin 3-O-ß-D-2″-acetylglucuronide and quercetin 3-O-ß-D-2″-acetylglucuronide methyl ester display π-π interactions by their phenyl moieties and Trp84 amino acid. Furthermore, another hydrogen bondings carried out by these three compounds also share some common residues to perform this type of interactions; however, there are some differences due to their accommodations into the enzyme pocket. In this manner, quercetin 3-O-ß-D-2″-acetylglucuronide showed five hydrogen bond interactions with Trp84, Tyr121, Tyr130 and His440 (already mentioned), quercetin 3-O-ß-D-2″-acetylglucuronide methyl ester exhibited six hydrogen bond interactions with Tyr121, Asp72, Tyr334, Gly441 and His440 and quercetin 3-O-ß-D-(glucuronide methyl ester) showed five hydrogen bondings with Gln 74, Tyr121, Ser122, Tyr130 and His440 (Figure 5D,E,G). Notwithstanding, quercetin 3-O-ß-D-2″-acetylglucuronide methyl ester exhibited the greatest protein–inhibitor complex stabilization, showing the best binding energy, suggesting that this compound could exert the main contribution to the inhibitory activity of *Ovidia pillopillo* leaves extract over the acetylcholinesterase.

Quercetin 3-O-(ß-D-glucuronide) maintain a similar orientation related to the three aforementioned compounds; nevertheless the 4*H*-chromen-4-one core of this latter derivative possess an opposite position, where the carbonyl function is inverted; therefore, it stabilizes within the catalytic site through hydrogen bond interactions with Gly118, Tyr121, Ser122 and His440, as well as by performing a π-π interaction with Phe288 and two T-shaped interactions with Phe331 and Trp233 (Figure 5F).

The toxic compound 8-O-methyl daphnin (daphnetin 7 O-(5″-O-methyl-glucose), 8-methyl ether) which is in the *Ovidia pillopillo* [1], showed a binding energy value of −12.651 kcal/mol suggesting that this compound could exert a good acetylcholinesterase inhibition. Into the catalytic site, compound 8-O-metyl daphnin performed three hydrogen bond interactions with Trp84, Tyr121 and Tyr130, as well as a T-shaped and a π-π interaction with Phe330 and Tyr334, respectively (Figure 5H).

#### 2.3.2. Butyrylcholinesterase (*h*BuChE) Docking Results

Binding energies from docking assays over butyrylcholinesterase (*h*BuChE) of the most abundant compounds from Lloime *Ovidia pillopillo* leaves extract showed to be less contributing compared to those obtained from acetylcholinesterase (*Tc*AChE). Nonetheless, all compounds presented a good binding energy profile, except for 5-hydroxy-7-methoxy-2-phenyl-4*H*-chromen-4-one, as in acetylcholinesterase (Table 3).

The compounds 5-hydroxy-7-methoxy-2-phenyl-4*H*-chromen-4-one and luteolin 7,4′-dimethyl ether displayed analogous poses into the butyrylcholinesterase catalytic site; nonetheless, they are not completely overlapped between each other, as in acetylcholinesterase experiments. Indeed, their structures are slightly tilted between them, where the phenyl ring of 5-hydroxy-7-methoxy-2-phenyl-4*H*-chromen-4-one is crooked in relation to the methoxyphenol ring of luteolin 7,4′-dimethyl ether. Given the above, 5-hydroxy-7-methoxy-2-phenyl-4*H*-chromen-4-one carried out one hydrogen bond interaction between the carbonyl group (C=O) of the 4*H*-chromen-4-one framework and Gly116, two T-shaped interactions between the 4*H*-chromen-4-one and the phenyl moiety with the residues of Trp82 and Phe329, respectively, and a π-cation interaction due the close distance between the imidazole ring of His438 and the 4*H*-chromen-4-one core. In the same way, Luteolin 7,4′-dimethyl ether performed a hydrogen bonding with ser198, a π-π interaction between Trp82 and the 4*H*-chromen-4-one, as well as π-π and π-cation interactions with the imidazole aromatic ring of His438 residue and the 4*H*-chromen-4-one mentioned above.

The same manner as in acetylcholinesterase docking assays, apigenin 5-glucoside binding pose into the butyrylcholinesterase catalytic site was completely different compared to all other tested derivatives, showing two hydrogen bond interactions through its two hydroxyl groups (-OH) at the glycoside moiety with Ser287 and two π-π interactions through the 4*H*-chromen-4-one framework with Trp82 and with the phenol ring, as well (Figure 6C).

Quercetin 3-O-ß-D-2″-acetylglucuronide and quercetin 3-O-ß-D-2″-acetylglucuronide methyl ester are almost spatially located in the same manner. In fact, both compounds perform hydrogen bond interactions through the oxygen atom of the hydroxyl groups at position 7- of the 4*H*-chromen-4-one and Tyr128 residue. Likewise, another hydrogen bond interaction through the hydrogen atom of the same already mentioned hydroxyl function of both derivatives are performed with Glu197. Besides, the 4*H*-chromen-4-one core of both compounds carry out a π-π interaction with Trp82. According to their differences, it can be seen a few of them between these two derivatives, for example a π-cation interaction performed by the phenolic ring of quercetin 3-O-ß-D-2″-acetylglucuronide and the amino acid His438, as well as three hydrogen bond interactions with Gly115 and Ala328 carried out by the appropriate hydroxyl functional groups of quercetin 3-O-ß-D-2″-acetylglucuronide (Figure 6D,E).

Docking experiments showed that quercetin 3-O-(ß-D-glucuronide) and quercetin 3-O-ß-D-(glucuronide methyl ester) are settled in a resemblant orientation into the butyrylcholinesterase catalytic site, but their structures are not exactly overlapping. The latter allows that both compounds can exhibit different docking results, where Quercetin 3-O-(ß -D-glucuronide) carry out a π-π interaction with Trp82 and owing it bears a carboxylate group in its glycoside moiety perform three strong hydrogen bond interactions with Gly116, Gly117 and His438, turning into the derivative with the best binding energy profile). Furthermore, two other same interactions with Tyr128 and Gly439 through both hydroxyl groups (-OH) of the 4*H*-chromen-4-one framework exist (Figure 6F). On the other hand, quercetin 3-O-ß-D-(glucuronide methyl ester) do not bear a carboxylate on its glycoside moiety; hence, it stabilizes into the butyrylcholinesterase catalytic site, performing hydrogen bond interactions through its hydroxyl functions (-OH) and the residues of Asp70, Thr120, Tyr128, Glu197 and His438, as well as through a π-π interaction between its 4*H*-chromen-4-one framework and Trp82 (Figure 6G).

Docking results for 8-O-metyl daphnin showed that this derivative fit into the butyrylcholinesterase in a different manner compared to the other tested compounds. This must be because 8-O-metyl daphnin correspond to a coumarin (2*H*-chromen-2-one); hence, its main structure as scaffold is different. The coumarin performed three hydrogen bond interactions: one between the hydroxymethyl group of the sugar moiety and Trp82 residue, and two hydrogen bondings with a hydroxy function of the sugar moiety and the amino acids Tyr 128 and Glu197. 8-O-metyl daphnin also performed a π-π interactions between the benzene ring of the coumarin and the indol ring of Trp82, contributing to the binding energy of −10.812 kcal/mol that this compound exhibited.

#### 2.3.3. Tyrosinase Docking Results

Binding energies from docking assays over tyrosinase of the most abundant compounds from *Ovidia pillopillo* leaves extract showed slightly better energies compared to kojic acid, that is to say, not a wide difference among the energy values was contemplated. In this sense, the extract did not exhibit a high inhibitory potency, reflecting this feature in the experimental IC_50_ obtained (9.92 ± 0.05 μg/mL). In terms of molecular interactions among each compound and the residues of the tyrosinase catalytic site, docking descriptors suggested that the main inhibitory activity would lie in quercetin 3-O-ß D-2″-acetylglucuronide and quercetin 3-O-(ß-D-glucuronide) derivatives. On the other hand, 5-hydroxy-7-methoxy-2-phenyl-4*H*-chromen-4-one docking assays over tyrosinase, just like in acetylcholinesterase and butyrylcholinesterase, showed a deficient binding energy of −5.908 kcal/mol and no contributing interactions were presented among the residues of the catalytic site and 5-hydroxy-7-methoxy-2-phenyl-4*H*-chromen-4-one structure, except for a π-π interaction with His263 and a hydrogen bond interaction with Met280 (Figure 7A). This phenomenon could be due to the lack of a glycoside core which bears many hydroxyl groups, ester or carboxylic functions capable of interacting with different amino acids. Thus, this compound would probably not contribute to the enzyme inhibition in a significant manner, even if it is in high proportion into the extract. In the same way, Luteolin 7,4′-dimethyl ether exhibited a better energy value of −6.054 kcal/mol. Although the 5-hydroxy-7-methoxy-2-phenyl-4*H*-chromen-4-one structure resembles to Luteolin 7,4′-dimethyl ether, this last one possesses, at its phenyl ring, a hydroxyl (-OH) function and a methoxy (-OCH_3_) group which, unlike 5-hydroxy-7-methoxy-2-phenyl-4*H*-chromen-4-one, allows one to perform hydrogen bond interactions and indeed executes them with Gly281 and Arg268, respectively. Furthermore, Luteolin 7,4′-dimethyl ether performed two π-π interactions with His263 and Phe264 (Figure 7B).

Apigenin 5-glucoside binding pose into the tyrosinase catalytic site once again did not match with the other derivatives assayed. However, this derivative performed three hydrogen bond interactions with Glu253, Asn260 and Val283, a T-shaped interaction with Phe264, as well as a π-cation between the phenyl moiety at position 2- of the 4*H*-chromen-4-one and Arg268 (Figure 7C).

Quercetin 3-O-ß-D-2″-acetylglucuronide and quercetin 3-O-ß-D-2″-acetylglucuronide methyl ester are arranged in a similar spatial manner; however, their 4*H*-chromen-4-one cores are in opposite positions. Due the above, both derivatives showed different interaction patterns with the enzyme. Indeed, Quercetin 3-O-ß-D-2″-acetylglucuronide showed three hydrogen bond interactions with His85, Val283 and Glu322, as well as a π-cation with Arg268, but also a salt bridge between one of the copper ions and the deprotonated carboxylate at its glycoside core, since the tyrosinase catalytic site possesses two copper cations in its morphology (Figure 7D). On the other hand, quercetin 3-O-ß-D-2″-acetylglucuronide methyl ester also performed hydrogen bond interactions with Glu256, Asn260, Gly281 and Glu322, as well as two π-π interactions with His85 and His244; however, instead of quercetin 3-O-ß-D-2″-acetylglucuronide, the derivative quercetin 3-O-ß-D-2″-acetylglucuronide methyl ester did not carry out any salt bridge because it lacks a carboxylate function and, as was already mentioned, because it is arranged in an opposite manner compared to quercetin 3-O-ß-D-2″-acetylglucuronide(Figure 7E).

Quercetin 3-O-(ß-D-glucuronide) and quercetin 3-O-ß-D-(glucuronide methyl ester) did not correlate in their binding poses, nor with any of the other derivatives. Nevertheless, it can be seen from Figure 7F,G that they share some interactions with the same amino acids, but with different organic functions of their structures. Therefore, quercetin 3-O-(ß-D-glucuronide) carried out five hydrogen bond interactions with, His244, Asn260, Arg268, His279, Gly281 and Val283, as well as a π- π interaction and a T-shaped with His263 and Phe264, respectively. Quercetin 3-O-ß-D-(glucuronide methyl ester) performed three hydrogen bond interactions with Glu256, Asn260 and Gly281, a π- π interaction with His263 and two t-shaped interactions with His259 and Phe264 residues (Figure 7G).

The compound 8-O-metyl daphnin, considering its binding energy value (−8.456 kcal/mol), could be expected to behave as a potent tyrosinase inhibitor. As in butyrylcholinesterase, this coumarin derivative is arranged in a different manner into the tyrosinase catalytic site in comparison to all other tested compounds. This last feature means that the main interactions performed by the 8-O-metyl daphnin were three hydrogen bond interactions, through the hydroxy groups of the sugar moiety and the amino acids His244, Glu256 and Asn260 and a π-π interaction performed between the 2*H*-chromen-2-one framework and Phe264.

Finally, in order to summarize the information, Figure 8 shows the main interactions in a two-dimensional diagram of the compound with the best binding energy profile of all derivatives found in high proportion within the *O. pillopillo* leaves extract and, consequently, the one that would contribute the most to the inhibitory activity. Each better compound, according to docking experiments, are shown into the catalytic sites of acetylcholinesterase, butyrylcholinesterase and tyrosinase enzymes, respectively.

## 3. Discussion

The use of plants rich in phenolic constituents has been important over the years to prevent neurodegenerative diseases due to the high content of polyphenolics. *The ethanolic extract of Lloime showed cholinesterase (*IC_50_ = 1.94 ± 0.07 and 2.73 ± 0.05 μg/mL, for AChE and BuChE, respectively) *and tyrosinase* (4.92 ± 0.05 μg/mL) enzyme inhibition activities and several detected flavonoids and coumarin are the probable responsible constituents for the reported activity. The plant also showed potent antioxidant capacity (Table 2) by several complementary assays. Regarding the metabolites identified in the ethanolic extract of O. pillopillo, some reports have indicated that, for example, the presence of coumarins such as daphnetin and daphnin are cholinesterase inhibitors helpful in Alzheimer’s disease [19], can also be active in the treatment of Parkinson’s disease [20], as well as cancer [21], and show other interesting activities in which oxidative stress may be involved. Those compounds were regarded as antibacterial, antifungal, antothrombotic, antiinflammatory, and can inhibit the enzymes cyclooxygenase, lipooxygenase and monoamine oxidase (MAO) [20]. Regarding the other compounds identified in the extract of this plant, glycosylated quercetin derivatives such as quercetin 3-O-glucuronide and isorahmnetin-3-O-glucuronide were reported to show acetylcholinesterase inhibitory activity, with IC_50_ values of 8.2 and 23.2 μM, respectively [22], those compounds also showed anti tyrosinase activity [23]. In this research,. we found cholinesterase and tyrosinase inhibitory activities of the ethanolic extract and all detected derivatives mentioned above showed good binding energies over acetylcholinesterase, even comparable to the reference compound galantamine in the docking calculations, including the reported toxic compound 8-O-methyl daphnin (daphnetin 7 O-(5″-O-methyl-glucose), 8-methyl ether), which showed a binding energy value of −12.651 kcal/mol suggesting that this compound could exert a good acetylcholinesterase inhibition. The ethanolic extract of this plant showed potential for the preparation of nutraceuticals or natural remedies.

## 4. Materials and Methods

### 4.1. Chemicals

Ultra-pure water (<5 µg/L TOC) was obtained from a water purification system Arium 126 61316-RO plus an Arium 611 UV unit (Sartorius, Goettingen, Germany). Methanol (HPLC grade) and formic acid (puriss. p.a. for mass spectrometry) were obtained from J. T. Baker (Phillipsburg, NJ, USA). Chloroform (HPLC grade) was obtained from Merck (Santiago, Chile). HPLC standards (Quercetin 3-O-(β-D-glucuronide, vanillic acid and chlorogenic acid, quercetin, acacetin, all standards with purity higher than 95% by HPLC) were obtained from Sigma-Aldrich Chem. Co. (St Louis, MO, USA) or Extrasynthèse (Genay, France). Gallic acid (purity > 98%), 6-hydroxy-2,5,7,8-tetramethylchromane-2-carboxylic acid (Trolox)(purity > 97%), 2,2′-azinobis(3-ethylbenothiazoline-6-sulfonic acid) diammonium salt (ABTS), 2,2-diphenyl-1-picrylhydrazyl (DPPH), Folin-Ciocalteu reagent 2,4,6-tri(2-pyridyl)-s-triazine (TPTZ), aluminum chloride, iron (III) chloride hexahydrate, (4-(2-hydroxyethyl)-1-piperazineethanesulfonic acid (HEPES), ethylenediaminetetraacetic acid (EDTA), adenosine 5′-triphosphate (ATP) disodium salt, acetylcholinesterase from Electrophorus electricus (electric eel) and butyrylcholinesterase from equine serum were from Sigma-Aldrich^®^ (Santiago, Chile). Calcium chloride, sodium carbonate, sodium hydroxide, sodium nitrite, and potassium persulphate were obtained from Merck^®^ (Chile). Acetone, galanthamine, kojic acid, zileuton, sodium acetate trihydrate, and glacial acetic acid were from Merck^®^ (Chile). HPLC reagents, formic acid, ethyl acetate, and n-hexane were form Merck^®^ (Chile).

### 4.2. Plant Material

Samples of the aerial parts (stems and leaves) of *Ovidia pillopillo* were collected during the months of September of the year 2018 in the capital city of Los Ríos Valdivia, close to being on the way to Oncol Park, Chile. The aerial parts were dried at room temperature and stored in the absence of light and heat. A voucher specimen was deposited in the ″Laboratorio de Productos Naturales of the Universidad Austral de Chile″ (voucher number OP-9-12-18).

### 4.3. Preparation of the Ethanolic Extract of O. pillopillo

The extract was prepared with dried and milled plant (stem and leaves) and absolute ethanol was employed, extracting 10 g of plant for 20 min with 50 mL ethanol (three times) with sonication. The extracts were combined, filtered, and concentrated in vacuo at 40 °C (yield 15%), and resulting gummy residue was stored at −80 °C for its further use.

### 4.4. Ultra-High Resolution Liquid Chromatography Orbitrap MS Analysis (UHPLC–PDA–OT-MS/MS) and Analysis Conditions

The analysis was performed with a UHPLC-high-resolution MS machine (Thermo Dionex Ultimate 3000 system with PDA detector controlled by Chromeleon 7.2 software hyphenated with a Thermo Q-exactive MS focus) with a RP-18 column (Thermo, Germany, 150 mm × 2.1 mm × 2,.5 μm particle size) at 25 °C. The detection wavelengths were 270, 250, and 330 nm, and photodiode array detectors set from 200 nm to 800 nm. Mobile phases were 1% formic aqueous solution (A) and acetonitrile 1% formic acid (B). The gradient program started at 5% B at time zero, was maintained at 7% B for 5 min, went to 40% B for 10 min, kept at 40% B for 15 min, went to 80% B for 5 min, kept at 80% B for 15 min and, finally, returned to the initial conditions within 15 min for column equilibration prior to each injection. The flow rate was 1.00 mL/min. The extract (5 mg) was dissolved in methanol (2 mL) and filtered through a 200 µm PTFE filter. Ten µL of this solution was injected in the instrument, considering all specifications (LC parameters and MS parameters) as reported [24]. Briefly, the parameters are as follows: sheath gas flow rate, 75 units; auxiliary gas unit flow rate, 20; capillary temperature, 400 °C; auxiliary gas heater temperature, 500 °C; spray voltage, 2500 V (for ESI−); and S lens, RF level 30. Full scan data in positive and negative modes were acquired at a resolving power of 70,000 FWHM at *m*/*z* 200. The mass scan range was between of 100–1000 *m*/*z*; automatic gain control (AGC) was set at 3 × 10^6^ and the injection time was set to 200 ms. The chromatographic system was coupled to MS with a source II heated electro-nebulization ionization probe (HESI II). The nitrogen gas carrier (purity > 99.999%) was obtained from a Genius NM32LA (Peak Scientific, Billerica, MA, USA) used as damping plus collision and gas. Mass calibration was performed once a day in both negative and positive modes to ensure working mass 5 ppm of accuracy. For the positive mode, a mixture of caffeine (1 mg/mL, 20 µL) and N-butylamine (1 mg/mL, 100 µL) was used, while a mixture of sodium dodecyl sulfate (1 mg/mL, 100 µL) and taurocholic acid sodium salt (1 mg/mL, 100 µL) (Sigma-Aldrich, Darmstadt, Germany) was used for the negative mode. In addition, Ultramark 1621 (Alpha Aezar, Stevensville, MI, USA) was used as the reference compound (1 mg/mL, 100 µL). These compounds were dissolved in a mixture of acetic acid (100 µL), acetonitrile (5 mL), water: methanol (1:1) (5 mL) (Merck, Santiago, Chile), and 20 µL of the mixture were infused using a Chemyx Fusion 100 µL syringe pump (Thermo Fisher Scientific, Bremen, Germany.

### 4.5. Determination of Total Phenolics and Flavonoids Content

The total content of phenolics and flavonoids compounds present in ethanolic extract of Lloime was determined by means of the Folin–Ciocalteu and AlCl_3_ tests respectively using a Synergy HTX microplate reader (Biotek, Winoosky, VT, USA) [25]. The evaluated concentrations of the extracts were 1 mg/mL (10 μL). The results were obtained with curves of standards (gallic acid (GA) and quercetin (Q), curves from 1 to 250 μg/mL, R^2^ = 0.99) and expressed as micrograms gallic acid equivalent per 100 g dry weight (μg GAE/100 g for phenolics compounds and micrograms quercetin equivalent per 100 g dry weight (μg QE/100 g dry weight) for flavonoids. The values from triplicates were reported as the mean ± SD.

### 4.6. Antioxidant Capacities of Lloime

Antioxidant capacities of extracts from Lloime were performed by FRAP, TEAC, ORAC and DPPH [25], each assay in triplicate.

#### 4.6.1. ABTS Assay

The ABTS free radical scavenging assay was performed as reported [25]. Briefly a solution of 7 mM ABTS (2,2′-azino-bis (3-ethylbenzothiazoline-6-sulphonic acid)) and a solution of 2.45 mM potassium persulfate in water were mixed in a 1:1 ratio (*v*/*v*) and incubated at room temperature for 16 h in the dark for the formation of the ABTS radical. After this period, a volume of 275 μL of solution was mixed thoroughly with 25 μL of standard or the samples, and the absorbance recorded at 734 nm using a microplate reader after 5 min. Gallic acid was used as a reference standard (from 1 to 100 μg/mL); a curve of extract was plotted (from 8 to 750 μg/mL). The results are expressed as IC_50_, in μg of the extract or standard per mL. The values are reported as the mean ± SD.

#### 4.6.2. DPPH Assay

DPPH free radical-scavenging activity was determined as previously reported [25]. A solution of DPPH 400 mM was prepared and using a volume of 150 μL of this solution thoroughly mixed with 50 μL of standard or the samples, the absorbance was recorded at 515 nm using a microplate reader and 96-well plates 30 min later. Gallic acid was used as a reference standard (from 1 to 100 mg. mL); curves of extracts of *O. pillopillo* (from 8 to 600 μg/mL) were prepared. The results are expressed as IC_50_, in μg of the extract or standard per mL. The values are reported as the mean ± SD.

#### 4.6.3. Ferric Reducing/Antioxidant Power the FRAP Assay

A previously validated method was employed [26]. Quantification was performed using a standard curve of the antioxidant Trolox (from 1 to 250 μg/mL, R2 = 0.99). The measurement was performed using a volume of 290 µL of extract in a well of the micro-plate reader and absorbance was measured at 593 nm after 5 min. The results were expressed in µmol of Trolox equivalents per g of dry plant. The experiments were performed in triplicates and the values are expressed as the mean ± SD.

#### 4.6.4. ORAC Assay

The ORAC assay was performed as previously described [26]. AAPH was used as peroxyl generator. The fluorescence of fluorescein disodium (FL) of each cycle was recorded. Parameters for the plate reader were: orbital shaking (4 mm shake width), cycle time, 210 s; position delay, 0.3 s; cycle number, 35; shaking mode, 8 s before each cycle; injection speed, 420 μL/s. Values were calculated by using a quadratic regression equation between the Trolox or sample concentration and net area under the FL decay curve. Data are expressed as micromoles of Trolox equivalents (TE) per gram of sample (μmol of TE/g).

The area under curve (AUC) was calculated as
AUC = (0:5 + f4/f3 + f5/f3 + f6/f3 + ⋯ + f32/f4 + f33/f4) × CT
where f3 = fluorescence reading at cycle 3; fn = fluorescence reading at cycle n; and CT, cycle time in minutes.

### 4.7. Cholinesterase Enzymes (ChE and BuCHe) Inhibition

The inhibitory activity of cholinesterase enzymes was measured using the Ellman’s method as previously reported [25]. Briefly, DTNB was dissolved in buffer Tris-HCl at pH 8.0 containing 0.1 M NaCl and 0.02 M MgCl_2_. Then, a filtered sample solution dissolved in deionized water (50 μL, 2 mg mL^−1^) was mixed with 125 μL of 5-dithio-bis(2-nitrobenzoic) acid (DTNB), acetylcholinesterase (AChE), or butyrylcholinesterase (BuChE) solution (25 μL) dissolved in Tris-HCl buffer at pH 8.0 in a 96-well microplate and incubated for 15 min at 25 °C. Initiation of reaction was performed by the addition of acetyl-thiocholine iodide (ATCI) or butyryl-thiocholine chloride (BTCl) (25 μL). In addition, a blank was prepared by adding the solution sample to all reagents without the enzyme(s) (AChE or BuChE) solutions and using as standard galantamine. The measurement was then performed at 405 nm, after 10 min of reaction. The activity was expressed as IC_50_ in μg/mL using a curve from 0.05 to 25 μg/mL.

### 4.8. Docking Simulations

Docking simulations were carried for compounds shown in Figure 4 and Figure 5; each one of them turned out to be the most characteristic and abundant species according to the UHPLC chromatogram (Figure 2) obtained from *O. pillopillo* leaves extract. First, the geometries and partial charges of every compound shown in Figure 4 were fully optimised using the DFT method with the standard basis set B3LYP-6-311G+ (d p) [27,28] in Gaussian 09W software [29]. Then, energetic minimizations and deprotonations were carried out using the LigPrep tool in Maestro Schrödinger suite v.11.8 (Schrödinger, LLC) [30]. Crystallographic enzyme structures of *Torpedo Californica* acetylcholinesterase (*Tc*AChE; PDBID: 1DX6 code [31]), human butyrylcholinesterase (*h*BuChE; PDBID: 4BDS code [32]) and the *Agaricus bisporus* mushroom tyrosinase (tyrosinase; PDBID: 2Y9X code [33]) were downloaded from the Protein Data Bank RCSB PDB [34]. Enzyme optimizations were carried out using the Protein Preparation Wizard available in Maestro software, where water molecules and ligands of the crystallographic protein active sites were removed. In the same way, all polar hydrogen atoms at pH = 7.4 were added. Appropriate ionization states for acid and basic amino acid residues were considered. The OPLS3e force field was used to minimize protein energy, as well. The enclosing box size was set to a cube with sides of 26 Å length. The centroid of selected residue were chosen based on the putative catalytic site of each enzyme, considering their known catalytic amino acids: Ser200 for acetylcholinesterase (*Tc*AChE) [35,36], Ser198 for butyrylcholinesterase (*h*BuChE) [37,38] and His263 for tyrosinase [33,39,40]. The Glide Induced Fit Docking protocol has been used for the final couplings [41]. Compounds were punctuated by the Glide scoring function in the extra-precision mode (Glide XP; Schrödinger, LLC) [42] and were filtered on the basis of the best scores and best RMS values (less than 1 unit as a cutting criterion), in order to obtain the potential intermolecular interactions between compounds and the enzymes, as well as the binding mode and docking descriptors. The different complexes were visualized in a visual molecular dynamics program (VMD) and Pymol [43].

### 4.9. Statistical Analysis

The data were statistically analyzed using the commercial software GraphPad Prism (GraphPad Software Inc., v.4, (San Diego, CA, USA), Statview (SAS Institute Inc., v5.0.1, Cary, NC, USA). For all the cases, the data were expressed as mean ± SEM and differences were considered significant at *p* < 0.05 and were determined by one-way ANOVA.

## 5. Conclusions

This work determined the chemical composition and antioxidant capacity of Lloime (*O. pillopillo*) ethanolic extract for the first time. Interesting toxic coumarins were identified; in addition, new glycosyl flavonoids showed interesting docking in the active sites of pharmacologically important enzymes, such as tyrosinase and cholinesterase. The results from the enzyme inhibition studies demonstrated a moderate inhibition, especially for the typical coumarins of the genera and glycosylated flavonoids, which was supported by the full docking experiments. More research is necessary in order to isolate the compounds and to perform more in vivo tests with the pure compounds that can support the beneficial and medicinal uses of this Valdivian Mapuche Amerindian plant.

## Figures and Tables

**Figure 1 metabolites-12-00090-f001:**
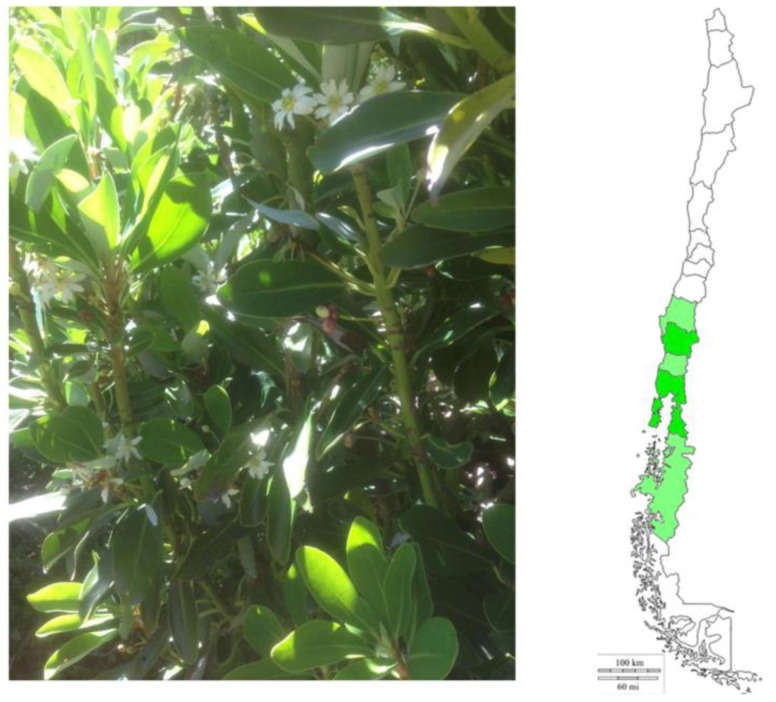
Picture of O. pillopillo collected at parque Oncol, Valdivia in November 2018 and zone of distribution in Chile.

**Figure 2 metabolites-12-00090-f002:**
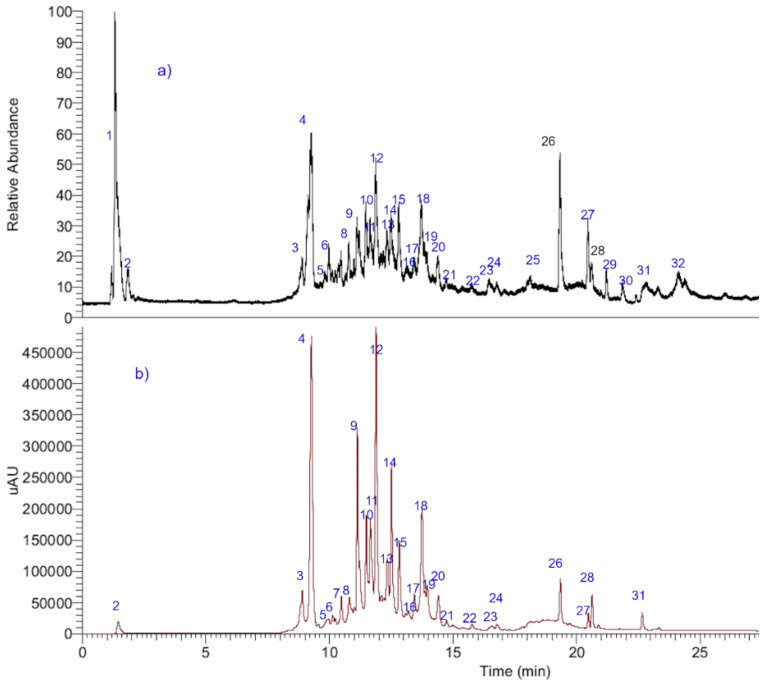
UHPLC chromatograms of Lloime (**a**) TIC, (**b**) UV chromatograms at 280 nm.

**Figure 3 metabolites-12-00090-f003:**
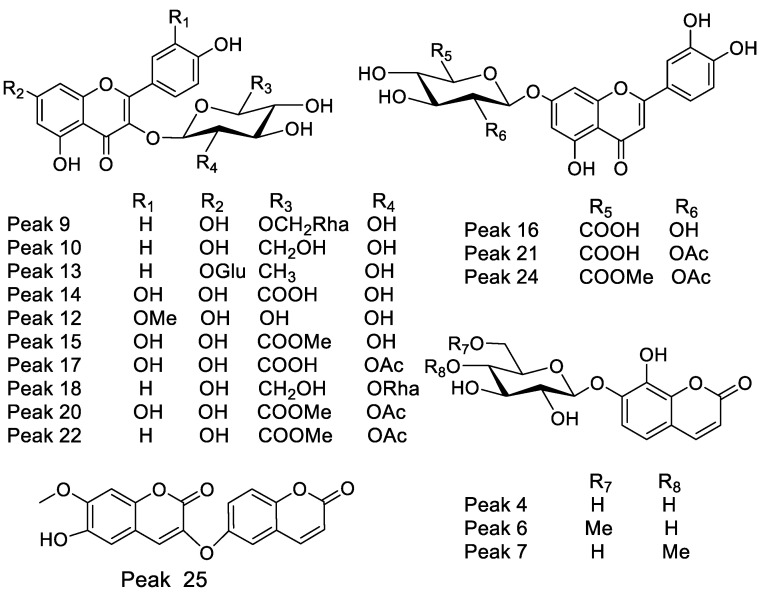
Main quercetin, kaempferol, luteolin and coumarin glycosyl derivatives detected in Lloime (*Ovidia pillopillo*).

**Figure 4 metabolites-12-00090-f004:**
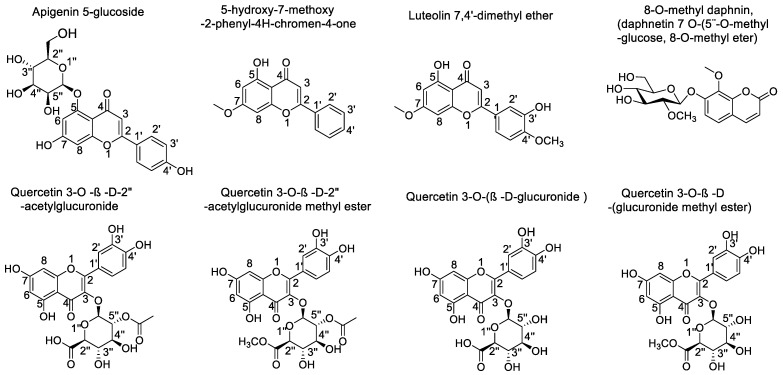
Main compounds in Lloime (Ovidia pillopillo) leaves used to perform docking experiments into the corresponding catalytic sites of acetylcholinesterase, butyrylcholinesterase, and tyrosinase.

**Figure 5 metabolites-12-00090-f005:**
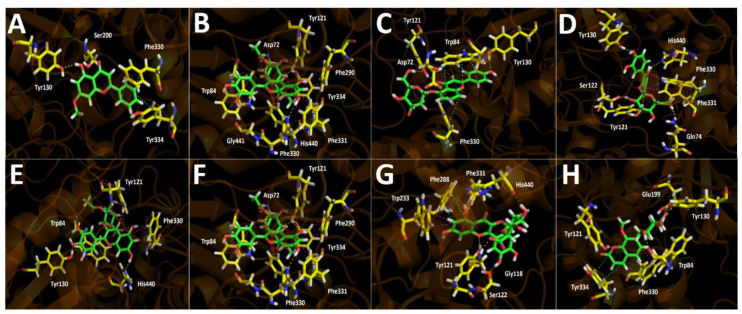
Predicted binding mode and predicted intermolecular interactions of most abundant compounds in *Ovidia pillopillo* leaves extract and the residues of *Torpedo Californica* acetylcholinesterase (*Tc*AChE) catalytic site. Yellow dotted lines indicate hydrogen bond interactions, cyan dotted lines represents π-π interactions, magenta dotted lines represents T-shaped interactions. (**A**). 5-hydroxy-7-methoxy-2-phenyl-4*H*-chromen-4-one into the catalytic site; (**B**). Luteolin 7,4′-dimethyl ether into the catalytic site; (**C**). Apigenin 5-glucoside into the catalytic site; (**D**). Quercetin 3-O-ß -D-2″-acetylglucuronide into the catalytic site; (**E**). Quercetin 3-O-ß-D-2″-acetylglucuronide methyl ester into the catalytic site; (**F**). Quercetin 3-O-(ß-D-glucuronide) into the catalytic site; (**G**). Quercetin 3-O-ß -D-(glucuronide methyl ester) into the catalytic site; (**H**). 8-O-metyl daphnin, (daphnetin 7 O-glucose, 8-methyl ether) into the catalytic site.

**Figure 6 metabolites-12-00090-f006:**
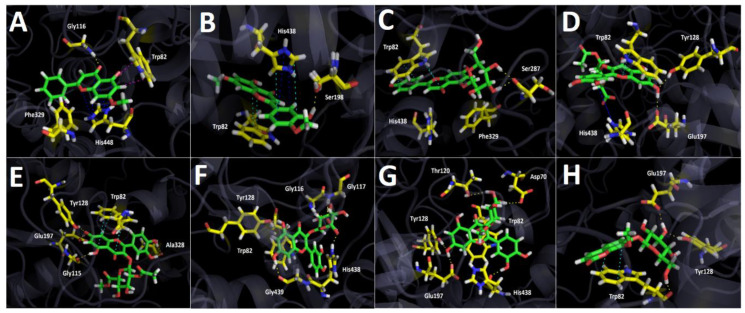
Predicted binding mode and predicted intermolecular interactions of most abundant compounds in *Ovidia pillopillo* leaves extract and the residues of *Torpedo Californica* acetylcholinesterase (*Tc*AChE) catalytic site. Yellow dotted lines indicates hydrogen bond interactions, cyan dotted lines represents π-π interactions, magenta dotted lines represents T-shaped interactions, blue dotted lines represents π -cation interactions. (**A**). 5-hydroxy-7-methoxy-2-phenyl-4*H*-chromen-4-one into the catalytic site; (**B**). Luteolin 7,4′-dimethyl ether into the catalytic site; (**C**). Apigenin 5-glucoside into the catalytic site; (**D**). Quercetin 3-O-ß -D-2″-acetylglucuronide into the catalytic site; (**E**). Quercetin 3-O-ß-D-2″-acetylglucuronide methyl ester into the catalytic site; (**F**). Quercetin 3-O-(ß-D-glucuronide) into the catalytic site; (**G**). Quercetin 3-O-ß -D-(glucuronide methyl ester) into the catalytic site; (**H**). 8-O-metyl daphnin, (daphnetin 7 O-glucose, 8-methyl ether) into the catalytic site.

**Figure 7 metabolites-12-00090-f007:**
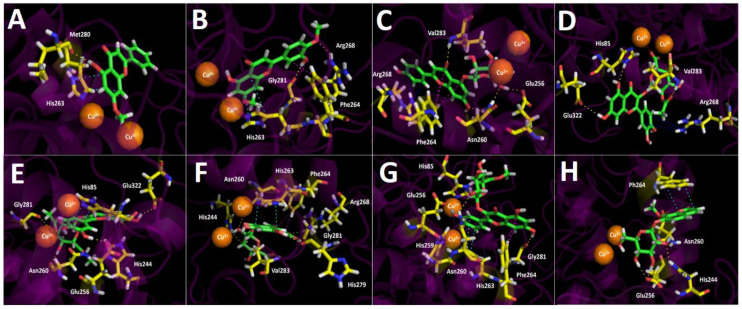
Predicted binding mode and predicted intermolecular interactions of most abundant compounds in *O. pillopillo* leaves extract and the residues of *Torpedo Californica* acetylcholinesterase (*Tc*AChE) catalytic site. Yellow dotted lines indicates hydrogen bond interactions, cyan dotted lines represents π-π interactions, magenta dotted lines represents T-shaped interactions, blue dotted lines represents π -cation interactions and red dotted lines represents salt bridges. (**A**). 5-hydroxy-7-methoxy-2-phenyl-4*H*-chromen-4-one into the catalytic site; (**B**). Luteolin 7,4′-dimethyl ether into the catalytic site; (**C**). Apigenin 5-glucoside into the catalytic site; (**D**). Quercetin 3-O-ß -D-2″-acetylglucuronide into the catalytic site; (**E**). Quercetin 3-O-ß-D-2″-acetylglucuronide methyl ester into the catalytic site; (**F**). Quercetin 3-O-(ß-D-glucuronide) into the catalytic site; (**G**). Quercetin 3-O-ß -D-(glucuronide methyl ester) into the catalytic site; (**H**). 8-O-metyl daphnin, (daphnetin 7 O-glucose, 8-methyl ether) into the catalytic site.

**Figure 8 metabolites-12-00090-f008:**
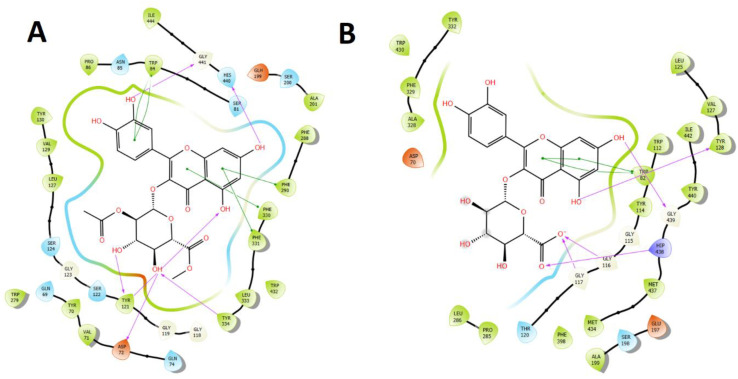
Two-dimensional diagram of (**A**)**.** Most active compound: Quercetin 3-O-ß-D-2″-acetylglucuronide methyl ester (related to docking binding energy) and its main interactions into the acetylcholinesterase (*Tc*AChE) catalytic site. (**B**)**.** Most active compound: Quercetin 3-O-(ß-D-glucuronide) (related to docking binding energy) and its main interactions into the butyrylcholinesterase (*h*BuChE) catalytic site. (**C**). Most active compound: Quercetin 3-O-ß D-2″-acetylglucuronide (related to docking binding energy) and its main interactions into the tyrosinase catalytic site. Purple arrows represents hydrogen bond interactions, green lines represents π-π interactions and red lines represents π-cation interactions.

**Table 1 metabolites-12-00090-t001:** UHPLC-PDA-MS identification of the ethanolic extract from Lloime (*Ovidia pillopillo*).

Peak #	Retention Time	UV Max	Tentative Identification	Molecular Formula [M-H]	Measured Mass (*m*/*z*)	Theoretical Mass (*m*/*z*)	Accuracy (ppm)	Ions MS^n^
1	1.45	227–272	Quinic acid	C_7_H_11_O_6__^−^_	191.05610	191.05610	1.385	175.06064
2	1.79	226	Isocitric acid	C_6_H_7_O_7__^−^_	191.02007	191.01917	−1.441	175.02426
3	8.90	227	Vanillic acid *	C_8_H_8_O_4__^−^_	168.04117	168.04225	−0.540	151.03951
4	8.95	227–290sh–325	Daphnin 8-O-methyl ether, (daphnetin 7 O-glucose, 8-methyl ether)	C_16_H_17_O_9__^−^_	353.08923	353.08779	2.521	191.05622 323.11307
5	9.26	228–260	Sinapoyl glucose	C_17_H_21_O_10__^−^_	385.11575	385.35066	2.287	341.06777, 223.05322
6	10.45	245–295sh–324	Daphnin 8-O-methyl ether-6″-O-methyl-glucose, (daphnetin 7 O-(6″-O-methyl-glucose, 8-O-methyl ether)	C_17_H_19_O_9__^−^_	367.10498	367.10345	2.621	323.11307, 191.05556, 175.03951
7	10.81	245–295sh 325	8-O-methyl daphnin, (daphnetin 7 O-(5″-O-methyl-glucose, 8-O-methyl ether)	C_17_H_19_O_9__^−^_	367.10492	367.10492	2.561	191.05620, 135.04460
8	10.87	255–355	Quercetin 3-O-xylosyl-(1-2)-rhamnoside	C_26_H_27_O_15__^−^_	579.13782	579.12446	3.374	301.03524
9	11.11	264–365	Kaempferol 3-O-rutinose	C_27_H_29_O_15__^−^_	593.15356	593.15356	5.83	449.10838, 285.03991
10	11.21	264–365	Kaempferol 3-O-β-D-glucose	C_21_H_19_O_11__^−^_	447.09503	447.09603	2.294	285.03991, 325.05595
11	11.40	277	Proanthocyanidin Geranin B	C_30_H_23_O_11__^−^_	559.12659	559.13186	5.548	255.06628, 541.11402
12	11.49	240–340	Isoscoparine	C_22_H_21_O_11__^−^_	461.10895	461.10894	1.112	299.05556, 283.02426
13	11.65	264–365	Kaempferitrin	C_27_H_29_O_14__^−^_	577.15786	577.15845	5.66	431.09837, 325.03537
14	11.88	254–354	Quercetin 3-O-(β-D-glucuronide) *	C_21_H_18_O_13__^−^_	477.06943	477.06691	3.063	301.03482, 433.07708
15	12.46	254–354	Quercetin 3-O-β-D-glucuronide- methyl ester	C_21_H_17_O_12__^−^_	491.08499	491.08470	6.054	301.03537
16	12.51	270–338	Luteolin-7-O-glucuronide	C_21_H_17_O_12__^−^_	461.07443	461.07254	5.269	285.03991, 267.02934
17	12.83	254–354	Quercetin-3-O-β-D(2″-acetyl glucuronide	C_23_H_19_O_14__^−^_	519.07990	519.07961	2.968	301.03529
18	13.45	265–365	Kaempferol-3-O-neohesperidose	C_27_H_25_O_5__^−^_	593.15596	593.15344	3.423	363.0770, 285.04843, 247.37323, 163.03951
19	13.75	280	3,8″-Binaringenin	C_30_H_23_O_10__^−^_	543.13165	543.13260	5.665	271.06119, 513.11910
20	13.91	254–354	Quercetin 3-O-β-D-(2″-O-acetyl- glucuronide methyl ester	C_24_H_21_O_14__^−^_	533.09556	533.01351	3.058	489.08342, 301.03412, 285.04843
21	13.92	265–355	Luteolin 7-O-(β-D-2′O-acetyl glucuronide	C_23_H_19_O_13__^−^_	503.08521	503.08521	3.058	491.08256, 315.05047, 285.03991
22	13.96	265–365	Kaempferol 3-O-β-(2′-O-acetyl-β-D-glucuronide	C_23_H_19_O_13__^−^_	503.08511	503.08310	3.093	285.03991, 461.07200, 459.09273
23	14.41	265–365	Kaempferol 3-O-glucuronide methyl ester	C_22_H_19_O_12__^−^_	475.09003	475.08856	2.928	285.03991, 429.08212
24	17.06	248–347	Luteolin 7-O-(β-D-2″O-acetylglucuronide methyl ester	C_24_H_21_O_13__^−^_	517.10034	517.10046	2.85	285.03991, 461.07200, 355.04539
25	19.33	227–290sh-325	Daphnoretin	C_19_H_12_O_7__^−^_	351.05252	351.05127	2.592	207.02995, 161.02441
26	19.33	265–365	Kaempferide (methyl kaempferol)	C_16_H_11_O_6__^−^_	299.05740	299.05556	2.385	273.03991, 151.00313, 147.04460
27	20.12	270–338	Quercetin *	C_15_H_9_O_7__^−^_	301.03513	301.03537	0.851	301.03427, 108.02057
28	20.46	267–335	Acacetin *	C_16_H_11_O_5__^−^_	283.06250	283.06065	1.852	273.03991, 257.05599
29	20.61	253–343	Hydroxyoctadecaenoic acid	C_18_H_35_O_3__^−^_	299.25907	299.25940	4.43	249.14966
30	20.63	253–343	3,5,7-Trihydroxyoleic acid	C_18_H_33_O_5__^−^_	329.23335	329.23346	3.12	119.04924 (C_8_H_7_O_^−^_)
31	22.65	268–330	Hexadecatrienoic acid	C_16_H_25_O_2__^−^_	249.17975	249.18600	−5.157	233.0
32	24.11	220	2-Hydroxypalmitate	C_16_H_31_O_3__^−^_	271.22677	271.22911	8.90	253.225

#: number, * Compounds identified by co spiking with authentic standards.

**Table 2 metabolites-12-00090-t002:** Scavenging of the 1,1-diphenyl-2-picrylhydrazyl Radical (DPPH), radical ABTS, (ABTS), Total phenolic content (TPC), Total flavonoid content (TFC) cholinesterase inhibition capacity and tyrosinase inhibition capacity of Lloime from the VIII Region of Chile. (n = 5).

Sample	DPPH ^a^	ABTS ^a^	ORAC ^b^	FRAP ^b^	TPC ^c^	TFC ^d^	AChE ^e^	BuChE ^e^	Tyr ^e^
Lloime ethanol extract	IC_50_ = 6.65 ± 0.5	IC_50_ = 9.95 ± 0.05	25.33 ± 1.2	45.56 ± 1.32	57.33 ± 0.82	38.42 ± 1.32	1.94 ± 0.07	2.73 ± 0.05	9.92 ± 0.05
Gallic acid	14.32 ± 0.5	1.67 ± 0.25	-	-	-	-	-	-	-
Kojic acid	-	-	-	-	-	-	-	-	3.51 ± 0.02
Galantamine	-	-	-	-	-	-	0.26 ± 0.02 ^e^	3.82 ± 0.02	-

^a^ Antiradical DPPH and ABTS activities are expressed as IC_50_ in μg/mL; ^b^ Expressed as μmol Trolox/g dry plant; ^c^ Total phenolic content (TPC) expressed as μg gallic acid equivalent GAE/100 g dry weight; ^d^ Total flavonoid content (TFC) expressed as μg equivalent of QE/100 g dry weight. ^e^ Cholinesterases and tyrosinase enzyme inhibitory activity in IC_50_ in μg/mL. Values in the same column are significantly different (at *p* < 0.05).

**Table 3 metabolites-12-00090-t003:** Binding energies obtained from docking experiments of most abundant compounds in *Lloime* leaves extract, as well as binding energies of the known inhibitors, galantamine and kojic acid over acetylcholinesterase (*Tc*AChE) butyrylcholinesterase (*h*BChE) and tyrosinase.

Compound	Binding Energy (kcal/mol) Acetylcholinesterase	Binding Energy (kcal/mol) Butyrylcholinesterase	Binding Energy (kcal/mol) Tyrosinase
5-hydroxy-7-methoxy-2-phenyl-4*H*-chromen-4-one	−9.154	−7.987	−5.908
Luteolin 7,4′-dimethyl ether	−10.506	−8.562	−6.054
Apigenin 5-glucoside	−12.798	−10.378	−9.018
Quercetin 3-O-ß-D-2″-acetylglucuronide	−14.064	−10.933	−10.333
Quercetin 3-O-ß-D-2″-acetylglucuronide methyl ester	−15.497	−11.803	−9.018
Quercetin 3-O-(ß-D-glucuronide)	−14.144	−12.888	−10.038
Quercetin 3-O-ß-D-(glucuronide methyl ester)	−14.518	−12.169	−9.169
8-O-methyl daphnin (daphnetin 7 O-(5″O-methyl-glucose, 8-methyl ether)	−12.651	−10.812	−8.456
Galantamine	−12.989	−7.125	-
Kojic acid	-	-	−6.050

## Data Availability

Data is contained within the article or Appendix A, but raw Thermo HPLC profiles of the plant or other data can be available on author’s request.

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
