# Peer review of "Phenolic Profile, Antioxidant and Enzyme Inhibition Properties of the Chilean Endemic Plant *Ovidia pillopillo* (Gay) Meissner (Thymelaeaceae)"

_metabolites, 2022, doi:10.3390/metabo12020090_

Round 1

Reviewer 1 Report

The presented manuscript shows a results of searching for compounds, which are present in Lloime plant, that may be applicable to developing functional foods.

The work is interesting, but it has a few shortcomings that should be supplemented or corrected.

In Abstract: “We have studied the antioxidant….” – it should be indicated why these properties are the basis for research.

What was the purpose of introducing the abbreviation "HRAM" if it was not used anywhere apart from the abstract?

The abstract and title show that the HPLC-MS method was used in the research, while the UPLC-MS technique is described in the text.

line 33: What does the term "characteristic coumarins" mean?

First part of Introduction: What healing properties do the mentioned parts of the plant show and in what form are they used in medicine? Are they registered pharmaceutical products or dietary supplements, or maybe products used in traditional medicine?

There is no information on the toxicity of individual parts of the plant in question.

Materials and Methods:

Have the procedures used for the determination of the described compounds been validated (in publications [20] and [21])? If so, it would be worth mentioning about it, presenting the most important parameters, and if not, it is necessary to carry out this process.

lines 484-486 and 663-664: The presented research may pave the way for further work on this plant as a source of active ingredients, while the statement that “…plant showed potential as supplement for …neurodegenerative diseases” is a bit premature. In order to be able to use this plant in the form of a dietary supplement, many more samples should be tested and various factors should be compared, i.e. the place of harvest, season, weather, etc. external factors that significantly affect the quality of the plant material.  

Ref [25]: Is it software or a link to a specific website?

Author Response

RESPONSE TO REVIEWER

RESPONSES ARE IN BLUE COLOR

The presented manuscript shows a results of searching for compounds, which are present in Lloime plant, that may be applicable to developing functional foods.

The work is interesting, but it has a few shortcomings that should be supplemented or corrected.

In Abstract: “We have studied the antioxidant….” – it should be indicated why these properties are the basis for research.

We have clarified this point

What was the purpose of introducing the abbreviation "HRAM" if it was not used anywhere apart from the abstract?

(HRAM), deleted

The abstract and title show that the HPLC-MS method was used in the research, while the UPLC-MS technique is described in the text.

changed

line 33: What does the term "characteristic coumarins" mean? Typical of the plant and genera

First part of Introduction: What healing properties do the mentioned parts of the plant show and in what form are they used in medicine? Are they registered pharmaceutical products or dietary supplements, or maybe products used in traditional medicine?

No registered nothing, Only in mapuche healing, causes vomiting, used to produce vomiting in intoxications and in low deposition conditions as strong purgative and to numb and catch fish.

There is no information on the toxicity of individual parts of the plant in question.

No information about this, this is interesting to perform for another paper

Materials and Methods:

Have the procedures used for the determination of the described compounds been validated (in publications [20] and [21])? If so, it would be worth mentioning about it, presenting the most important parameters, and if not, it is necessary to carry out this process.

Added as requested.

lines 484-486 and 663-664: The presented research may pave the way for further work on this plant as a source of active ingredients, while the statement that “…plant showed potential as supplement for …neurodegenerative diseases” is a bit premature. In order to be able to use this plant in the form of a dietary supplement, many more samples should be tested and various factors should be compared, i.e. the place of harvest, season, weather, etc. external factors that significantly affect the quality of the plant material.  

Ok deleted the phrase

The results from the enzyme inhibition studies demonstrated a moderate inhibition, specially for the typical coumarins of the genera and glycosylated flavonoids, supported by the full docking experiments. More research is necessary, to isolate the compounds and to perform more in vivo tests with the pure compounds that can support the beneficial and medicinal uses of the Valdivian plant.

Ref [25]: Is it software or a link to a specific website?

A software Gaussian

Frisch, A. Gaussian 09W Reference; Gaussian: Wallingford, UK, 2009

Reviewer 2

The article entitled “Ovidia pillo‐pillo (Gay) Meissner (Thymelaeaceae), HPLC‐MS Phenolic Profile, Antioxidant and Anticholinesterase Enzyme Inhibition Properties” deals with the antioxidant and enzyme inhibition activities. Moreover, the phenolic profile was analyzed by UHPLC-PDA-OT‐MS/MS. The investigation results are interesting; however, the article is prepared carelessly. I strongly recommend substantial language editing before acceptance.

I have the following comments for this article:

The article has numerous languages, grammatic, typological issues; here I am addressing the few:

  • The plant name “Ovidia pillo‐pillo (Gay) Meissner” is correct? Wikipedia says, “Ovidia pillopillo (Gay) Meisn.”. Moreover, in line 661 it is written as “ pillo pillo
  • In the article “Ovidia pillo‐pillo” is written inconsistently, for instance, lines 26, 81, 85, 93, and many other places (Lloim; in italics); lines 43, 73, 106, and many other places (Lloim; no italics)
  • changed
  • Punctuation errors: Line 44 “Valdivia, (Figure 1) the bark, fruits”; Line 106 “Lloime, ‐O. Pillo‐Pillo‐ (Table 1)”; Line 114 “as daphnin 8‐O‐methyleter, (daphnetin 7 O‐glucose, 8‐methyl eter, C16H17O9‐)”; Line 134 “isomer, peak 18, as kaempferol‐3‐O‐neohesperidose”; Line 222 “kcal/mol respectively)”; Line 610 “previously reported [21] . Briefly”
  • CHECKED
  • Typological errors: Line 112, 114, 115, and several places in the article “eter”
  • Unnecessary capitalization of the first letter: line 121 (Sinapopyl), line 149 (Hydroxyoctadecaenoic), line 150 (Hexadecatrienoic and Hydroxypalmitate), line 210 (Galantamine), line 223-224 (Quercetin), and many other words throughout the article done
  • Inconsistent use of abbreviations: for instance, the use of abbreviations acetylcholinesterase (AChE) and Butyrylcholinesterase (BChE) were not followed properly done
  • Long sentences in the parenthesis: Line 495-496 and 531-533 done
  • Please revise the citation style: Line 568 “determined according to [21]” done
  • Line 90: “UHPLC‐OT.MS” should be written as “UHPLC-OT-MS” or “UHPLC-OT-MS/MS” done

 REVIEWER 2

Table 2: Values in the same column are significantly different (at p < 0.05); Two way ANOVA not found in the table as mentioned in line 658 corrected

Line 468-473: This sentence needs rewriting

done

4.1. Chemicals: the author used “ultra‐pure water” (Line 491) or “deionized water” (Line 509)?. Moreover, HPLC methanol is written two times (line 493 and line 508)

Sorry, corrected

Line 558: Incubation time? done

Line 582-583: incomplete sentence with several other typological errors

changed

Line 657: “data were expressed as mean ± SEM”, in contrast, line 545 says “values from triplicates were reported as the mean ± SD” changed

Conclusions: needs rewriting

changed

REVIEWER 3

I read carefully your manuscript sent for possible publication in Metabolites.

I want to mention that the laboratory work done it by your team is huge, the results are well presented, and the figures included are explanatory.

From my point of view the manuscript can be take into consideration for publication after some minor spelling errors correction and maybe is appropriate to insert some others scientific paper regarding the main HPLC‐MS, Phenolic Profile, Antioxidant and Anticholinesterase Enzyme Inhibition Properties. 

Best regards!

thanks

REVIEWER 4

Introduction is poor and chaotic. Authors should highlighted why such activity was investigated. The significance of the study should be also emphasised.  Some information are unnecessary, e.g. lines 75-77 – the advantages of UHPLC method and its utility  is commonly known

rewrited

Title of the manuscript should be rewritten. Changed

Phenolic Profile, Antioxidant and Enzyme Inhibition Properties of the Chilean Endemic plant Ovidia pillopillo (Gay) Meissner (Thymelaeaceae).

Lines 85-87 are unnecessary because this information is in Material and Methods

deleted

2.1. Section should be reedited: lines 93-94 are unnecessary. Information in section 2.1.1 -2.1.4 are unnecessary because the data are in Table 1. DELETED

We have added some more references and comments.this is typical way of commenting this part.

The description of figure should be placed under the figure done

Conclusion: statement: „The plant showed potential as supplement for the palliative use for neurodegenerative diseases” is weird because the plant contain a lot of toxic compounds.

Toxic compounds could be also drugs, such as coumarins, anyway  as reviewer requested, we delete that phrase.

Replace “Tincture” by “extract” done

“as μg gallic acid quercetin/trolox/100 g” – the word  “equivalent” should be added. Check all text. done

4.3.1. Section: more detail on UHPLC analysis should be added (type of column, mobile phase…) done

Line 461: “The use of plants rich in phenolic constituents has been important over the years to prevent neurodegenerative diseases due to the high content of polyphenolics”. – delate the second part of the sentence (due to….) yes

Line 582: “a buffer solution CH3Na*3 H2O 3.1%/CH3COOH (glacial) 16%, plus 20 mM FeCl3 in aqueous solution HCl 0.02 M and 10 mM TPTZ dissolved in absolute ethanol.” – reedit the sentence. “CH3Na*3 H2O 3.1%/CH3COOH (glacial) 16%” – is completely nuclear deleted

The formulas for “% bleaching” are unnecessary because the % values are not shown in manuscript deleted

There are a lot of minor errors, e.g. „ultrahigh”, „in vitro” should be in italic, lack of space (e.g. line 464, 553 ..), lack of capital letters for ml (e.g. 4.3.1. section) done

Reviewer 2 Report

The article entitled “Ovidia pillo‐pillo (Gay) Meissner (Thymelaeaceae), HPLC‐MS Phenolic Profile, Antioxidant and Anticholinesterase Enzyme Inhibition Properties” deals with the antioxidant and enzyme inhibition activities. Moreover, the phenolic profile was analyzed by UHPLC-PDA-OT‐MS/MS. The investigation results are interesting; however, the article is prepared carelessly. I strongly recommend substantial language editing before acceptance.

I have the following comments for this article:

The article has numerous languages, grammatic, typological issues; here I am addressing the few:

  • The plant name “Ovidia pillo‐pillo (Gay) Meissner” is correct? Wikipedia says, “Ovidia pillopillo (Gay) Meisn.”. Moreover, in line 661 it is written as “ pillo pillo
  • In the article “Ovidia pillo‐pillo” is written inconsistently, for instance, lines 26, 81, 85, 93, and many other places (Lloim; in italics); lines 43, 73, 106, and many other places (Lloim; no italics)
  • Punctuation errors: Line 44 “Valdivia, (Figure 1) the bark, fruits”; Line 106 “Lloime, ‐O. Pillo‐Pillo‐ (Table 1)”; Line 114 “as daphnin 8‐O‐methyleter, (daphnetin 7 O‐glucose, 8‐methyl eter, C16H17O9‐)”; Line 134 “isomer, peak 18, as kaempferol‐3‐O‐neohesperidose”; Line 222 “kcal/mol respectively)”; Line 610 “previously reported [21] . Briefly”
  • Typological errors: Line 112, 114, 115, and several places in the article “eter”
  • Unnecessary capitalization of the first letter: line 121 (Sinapopyl), line 149 (Hydroxyoctadecaenoic), line 150 (Hexadecatrienoic and Hydroxypalmitate), line 210 (Galantamine), line 223-224 (Quercetin), and many other words throughout the article
  • Inconsistent use of abbreviations: for instance, the use of abbreviations acetylcholinesterase (AChE) and Butyrylcholinesterase (BChE) were not followed properly
  • Long sentences in the parenthesis: Line 495-496 and 531-533
  • Please revise the citation style: Line 568 “determined according to [21]”
  • Line 90: “UHPLC‐OT.MS” should be written as “UHPLC-OT-MS” or “UHPLC-OT-MS/MS”

Table 2: Values in the same column are significantly different (at p < 0.05); Two way ANOVA not found in the table as mentioned in line 658

Line 468-473: This sentence needs rewriting

4.1. Chemicals: the author used “ultra‐pure water” (Line 491) or “deionized water” (Line 509)?. Moreover, HPLC methanol is written two times (line 493 and line 508)

Line 558: Incubation time?

Line 582-583: incomplete sentence with several other typological errors

Line 657: “data were expressed as mean ± SEM”, in contrast, line 545 says “values from triplicates were reported as the mean ± SD”

Conclusions: needs rewriting

Author Response

(The authors gave the same response as above.)

Reviewer 3 Report

Dear Authors, 

I read carefully your manuscript sent for possible publication in Metabolites.

I want to mention that the laboratory work done it by your team is huge, the results are well presented, and the figures included are explanatory.

From my point of view the manuscript can be take into consideration for publication after some minor spelling errors correction and maybe is appropriate to insert some others scientific paper regarding the main HPLC‐MS, Phenolic Profile, Antioxidant and Anticholinesterase Enzyme Inhibition Properties. 

Best regards!

Author Response

(The authors gave the same response as above.)

Reviewer 4 Report

Introduction is poor and chaotic. Authors should highlighted why such activity was investigated. The significance of the study should be also emphasised.  Some information are unnecessary, e.g. lines 75-77 – the advantages of UHPLC method and its utility  is commonly known

Title of the manuscript should be rewritten.

Lines 85-87 are unnecessary because this information is in Material and Methods

2.1. Section should be reedited: lines 93-94 are unnecessary. Information in section 2.1.1 -2.1.4 are unnecessary because the data are in Table 1.

The description of figure should be placed under the figure

Conclusion: statement: „The plant showed potential as supplement for the palliative use for neurodegenerative diseases” is weird because the plant contain a lot of toxic compounds.

Replace “Tincture” by “extract”

“as μg gallic acid quercetin/trolox/100 g” – the word  “equivalent” should be added. Check all text.

4.3.1. Section: more detail on UHPLC analysis should be added (type of column, mobile phase…)

Line 461: “The use of plants rich in phenolic constituents has been important over the years to prevent neurodegenerative diseases due to the high content of polyphenolics”. – delate the second part of the sentence (due to….)

Line 582: “a buffer solution CH3Na*3 H2O 3.1%/CH3COOH (glacial) 16%, plus 20 mM FeCl3 in aqueous solution HCl 0.02 M and 10 mM TPTZ dissolved in absolute ethanol.” – reedit the sentence. “CH3Na*3 H2O 3.1%/CH3COOH (glacial) 16%” – is completely unclear

The formulas for “% bleaching” are unnecessary because the % values are not shown in manuscript

There are a lot of minor errors, e.g. „ultrahigh”, „in vitro” should be in italic, lack of space (e.g. line 464, 553 ..), lack of capital letters for ml (e.g. 4.3.1. section)

Author Response

(The authors gave the same response as above.)

Round 2

Reviewer 2 Report

The authors have not considered the comments critically. Moreover, no proper repose is provided for the comments. A response is like “Sorry, corrected” is appropriate for a high-quality journal?.

The authors used the copyrighted material in figure1. Figure 1, which authors say was taken in 2018, is actually taken on November 28, 2009, https://chilebosque-cl.appspot.com/?name=Ovidia+pillopillo.

The article still contains several typological errors. For instance, the first sentence of the introduction itself is wrong, and despite the previous comments, this is not corrected.

Previously, I strongly recommended substantial language editing before acceptance. However, the authors have not considered these comments (no response provided)

Line 77: UHPLC stands for “Ultra-high-resolution chromatography”?

Line 125-127: punctuation and typological errors in the sentence “peak 18 as its acetyl derivative quercetin-3-O-β-D (2’’-acetyl glucuronide (C23H19O14-), and peak 16 as its methyl derivative quercetin 3-O-β-D-glucuronide-methyl esther (C21H17O12-) and peak 21 as quercetin 3-O-(2”-O-acetyl-glucuronide methyl ester), with.” The “Esther” used in this sentence is correct?.

Author Response

Reviewer 2

The authors have not considered the comments critically. Moreover, no proper repose is provided for the comments. A response is like “Sorry, corrected” is appropriate for a high-quality journal?.

Sorry dear reviewer we will answer more appropriately to you. Thank you for all.

The authors used the copyrighted material in figure1. Figure 1, which authors say was taken in 2018, is actually taken on November 28, 2009, https://chilebosque-cl.appspot.com/?name=Ovidia+pillopillo.

Thank you to the dear reviewer for the corrections. Sorry, my student provided the picture I thought it was his picture. We changed the picture for one taken by our group in 2018 as indicated.

The article still contains several typological errors. For instance, the first sentence of the introduction itself is wrong, and despite the previous comments, this is not corrected.

Thank you we rewrote the introduction to make it more appropriate.

Previously, I strongly recommended substantial language editing before acceptance. However, the authors have not considered these comments (no response provided)

We did our best to improve language, sorry for that.

Line 77: UHPLC stands for “Ultra-high-resolution chromatography”?

Dear reviewer, UHPLC means Ultra-high resolution liquid chromatography.we corrected the phrase.

Line 125-127: punctuation and typological errors in the sentence “peak 18 as its acetyl derivative quercetin-3-O-β-D (2’’-acetyl glucuronide (C23H19O14-), and peak 16 as its methyl derivative quercetin 3-O-β-D-glucuronide-methyl esther (C21H17O12-) and peak 21 as quercetin 3-O-(2”-O-acetyl-glucuronide methyl ester), with.” The “Esther” used in this sentence is correct?.

Thank you to the dear reviewer for the corrections. Sorry it is a mistake, we corrected.

Reviewer 4 Report

My intention was to point out the weaknesses of the manuscript to allow make it better. Unfortunately, the authors did not incorporate the majority of suggested improvements (although they declared in response that they did it). They even did not include  the explanation for reviewer why they consider them as redundant.

Abstract: “ eurodegenerative diseases” ?

Introduction was not improved. The significance of the study was not emphasized.  Lines 75-77 was not removed (the advantages of UHPLC method and its utility are commonly known)

Line 50: “toxic 7,8-hydroxycoumarin daphnin: daphnetin 7-β-D-glucopyranoside [1],” –the cited paper was from 1945. Was this compound identified at 1945?  Ref. 9–13 are not directly linked with the study – justify why was necessary to cited these papers (self-citation not linked with the work should be avoid)

Line 91-93: “(…) revealed the presence of thirty two compounds in Lloime, (…). The composition of the  extracts included phenolics acids, fatty acids, and several characteristic toxic coumarins” and line 103-105: “ (…) the presence of thirty-two compounds in the metabolome of Lloime (Table 1). Among these compounds (Figure 3) were four coumarins, two phenolic acids, twenty flavonoids and four fatty acids.” contain similar information. Reedit this phrase

Regarding 2.1.1 – 2.1.4.   Authors response: It is a typical way of commenting this part...

Comment: It is valuable way for describing new compounds; however, in this paper identification was done based on literature and all compounds were previously reported. Thus, this part only replicated the data in Table.  If appropriate references would be added to the Table this part is unnecessary. Currently, there is a tendency to present the investigation in compact form

“as μg gallic acid quercetin/trolox/100 g” – the word  “equivalent” should be added. Check all text. – it was not corrected (see e.g. line 173, 175, 180… Abstract)

Line 535: „2,5 mM particle size”? „a solvent system of” – of what?

Why was the section „Determination of total phenolics and flavonoids content” included as a part of 4.4. Ultrahigh resolution liquid chromatography Orbitrap MS analysis? TPC/TFC are spectrophotometric assay.

„4.3.1 UHPLC-DAD-MS Instrument” – reedit the title. This section also contains the analysis conditions.

Line 595, 608 – lack of numbering for section DPPH,  „Ferric Reducing/Antioxidant Power the FRAP Assay”

The formulas for “% bleaching” are unnecessary because the % values are not shown in manuscript – it was not corrected. Why did Authors decide to keep these formulas?

Ultrahigh – it should be “ultra-high”

Correct references – lack of italic, lack of capital letters (e.g. ref. 9)

Author Response

Reviewr 4

My intention was to point out the weaknesses of the manuscript to allow make it better.

Dear reviewer, thank you very much for your feedback which we believe will enhance the work presented. Below are the answers to your comments:

Unfortunately, the authors did not incorporate the majority of suggested improvements (although they declared in response that they did it). They even did not include  the explanation for reviewer why they consider them as redundant.

We regret that we did not provide adequate responses to your comments. We have done our best to improve the writing.

Abstract: “ eurodegenerative diseases” ?

Thank you to the dear reviewer for the corrections. We have corrected.

Introduction was not improved. The significance of the study was not emphasized.  Lines 75-77 was not removed (the advantages of UHPLC method and its utility are commonly known)

Thank you we rewrote the entire introduction to make it more appropriate. We removed the advantages of UHPLC method.

Line 50: “toxic 7,8-hydroxycoumarin daphnetin 7-β-D-glucopyranoside [1],” –the cited paper was from 1945. Was this compound identified at 1945?  

Dear reviewer, we do not know how the identification of the compound was achieved in 1945, however subsequent studies support the presence of this compound in the plant:

Zhang, T.; Wei, F.; Liu, S.; Zhao, S.; Liu, L.; Zhao, H.; Li, Y.; Zhang, T.; Wei, F.; Liu, S.; et al. Identification and Quantification of Chemical Constituents in Daphne altaica and their Antioxidant and Cytotoxic Activities. Int. J. Agric. Biol. 2019, doi:10.17957/IJAB/15.1159.

Deiana, M.; Rosa, A.; Casu, V.; Cottiglia, F.; Bonsignore, L.; Dessì, M.A. Chemical composition and antioxidant activity of extracts from Daphne gnidium L. JAOCS, J. Am. Oil Chem. Soc. 2003, 80, 65–70, doi:10.1007/s11746-003-0652-x.

Ref. 9–13 are not directly linked with the study – justify why was necessary to cited these papers (self-citation not linked with the work should be avoid)

Dear reviewer, we emphasize with these references since our research group aims to study native medicinal plants and this work is a continuation of previous works in the search for "medicinal" plants or their potential.

Line 91-93: “(…) revealed the presence of thirty two compounds in Lloime, (…). The composition of the  extracts included phenolics acids, fatty acids, and several characteristic toxic coumarins” and line 103-105: “ (…) the presence of thirty-two compounds in the metabolome of Lloime (Table 1). Among these compounds (Figure 3) were four coumarins, two phenolic acids, twenty flavonoids and four fatty acids.” contain similar information. Reedit this phrase

Dear reviewer, this part was rewritten thanks.

Regarding 2.1.1 – 2.1.4.   Authors response: It is a typical way of commenting this part...

Comment: It is valuable way for describing new compounds; however, in this paper identification was done based on literature and all compounds were previously reported. Thus, this part only replicated the data in Table.  If appropriate references would be added to the Table this part is unnecessary. Currently, there is a tendency to present the investigation in compact form

Dear reviewer, thank you for you feedback but we believe that this part should not be eliminated from the writing since, although the identified compounds are known, the analysis that allows their identification is relevant.

“as μg gallic acid quercetin/trolox/100 g” – the word  “equivalent” should be added. Check all text. – it was not corrected (see e.g. line 173, 175, 180… Abstract)

Dear reviewer, we check this.

Line 535: „2,5 mM particle size”? „a solvent system of” – of what?

Dear reviewer, we removed this sentence, as it was incorrect, thank you.

Why was the section „Determination of total phenolics and flavonoids content” included as a part of 4.4. Ultrahigh resolution liquid chromatography Orbitrap MS analysis? TPC/TFC are spectrophotometric assay.

Dear reviewer this was an error, it is already corrected thanks.

„4.3.1 UHPLC-DAD-MS Instrument” – reedit the title. This section also contains the analysis conditions.

Thank you to the dear reviewer for the corrections. We have reedit the title and added conditions

Line 595, 608 – lack of numbering for section DPPH, Ferric Reducing/Antioxidant Power the FRAP Assay”

Dear reviewer, we corrected the corresponding numbers, thank you

The formulas for “% bleaching” are unnecessary because the % values are not shown in manuscript – it was not corrected. Why did Authors decide to keep these formulas?

Sorry, dear reviewer we have deleted the formulas

Ultrahigh – it should be “ultra-high”

Thanks for the clarification. This has been changed in the text.

Correct references – lack of italic, lack of capital letters (e.g. ref. 9)

Dear reviewer, we corrected the references.

Round 3

Reviewer 2 Report

NA

Author Response

Responses are in blue

Reviewer 2

Open Review

English language and style

( ) Extensive editing of English language and style required
(x) Moderate English changes required
( ) English language and style are fine/minor spell check required
( ) I don't feel qualified to judge about the English language and style

Yes

Can be improved

Must be improved

Not applicable

Does the introduction provide sufficient background and include all relevant references?

( )

(x)

( )

( )

Is the research design appropriate?

(x)

( )

( )

( )

Are the methods adequately described?

( )

(x)

( )

( )

Are the results clearly presented?

( )

(x)

( )

( )

Are the conclusions supported by the results?

( )

(x)

( )

( )

Comments and Suggestions for Authors

NA

English was revised again

Submission Date

06 December 2021

Date of this review

12 Jan 2022 11:08:57

Reviewer 4

Dear Authors, Thank You for your kind explanation and detailed responses. I really appreciate the efforts made for investigation of new endemic species. Although the changes in Introduction are rather cosmetic, I accept all corrections and explanation. However, there are still minor errors in text.

Lines 31,33 (Abstract), 159,161, 166 : add the word “equivalent”

Thank you to the reviewer for the careful corrections, it was done

Title of section 4.4.:  “instrument conditions” – the expression: “analysis conditions” is more adequate.

Thank you to the reviewer for the careful corrections, it was done

Line 508: „2,5 mM particle size” – mM (milimol) is not a units for particle size (probably it should be “µm”). “ 2 cm diameter” – is it correct? (probably it should be 2 mm).

Thank you to the reviewer for the careful corrections, it was done

Line 541: “The results were obtained with curves of standards (gallic acid GAE and quercetin QE, curves” – the abbreviation GAE/QE are typically used for expression “gallic acid/quercetin equivalent”. Calibration curves were prepared using gallic acid (GA) and quercetin (Q)

Thank you to the reviewer for the careful corrections, it was done

Reviewer 4 Report

Dear Authors, Thank You for your kind explanation and detailed responses. I really appreciate the efforts made for investigation of new endemic species. Although the changes in Introduction are rather cosmetic, I accept all corrections and explanation. However, there are still minor errors in text.

Lines 31,33 (Abstract), 159,161, 166 : add the word “equivalent”

Title of section 4.4.:  “instrument conditions” – the expression: “analysis conditions” is more adequate.

Line 508: „2,5 mM particle size” – mM (milimol) is not a units for particle size (probably it should be “µm”). “ 2 cm diameter” – is it correct? (probably it should be 2 mm).

Line 541: “The results were obtained with curves of standards (gallic acid GAE and quercetin QE, curves” – the abbreviation GAE/QE are typically used for expression “gallic acid/quercetin equivalent”. Calibration curves were prepared using gallic acid (GA) and quercetin (Q)

Author Response

(The authors gave the same response as above.)
